# Mind the Gap: Transferring Labels Across Datasets with Divergent Annotation Protocols

## Abstract

Combining multiple object detection datasets offers a promising path to improved model generalisation. However, inconsistencies in class semantics and bounding box annotations present significant challenges. Most existing approaches either assume shared label taxonomies, address only spatial inconsistencies, or require manual relabeling, limiting their scalability and flexibility. We propose Label-Aligned Transfer (LAT), a framework that systematically projects annotations from diverse source datasets into the label space of a target dataset. LAT begins by training dataset-specific detectors to generate pseudo-labels. These pseudo-labels are then fused with ground-truth annotations via a Privileged Proposal Generator (PPG), which replaces the region proposal network in two-stage detectors to build shared proposals across datasets. To refine region features and mitigate pseudo-label noise, a Semantic Feature Fusion (SFF) module applies class-aware attention, aligning features while filtering unreliable signals. Unlike ontology harmonisation or embedding-based unification, LAT operates directly at the region and feature levels, avoiding semantic drift and preserving fine-grained dataset-specific granularity. It supports many-to-one label projection without requiring shared label spaces or manual reannotation, enabling effective training over heterogeneous corpora. Empirical evaluations across multiple benchmark combinations show consistent performance gains, with improvements of up to +8.4 AP over competitive baselines.

## 1 Introduction

Combining multiple datasets has become an increasingly practical strategy for improving object detection performance, particularly in domains where annotated data is scarce or expensive. However, naively merging datasets with differing label spaces introduces inconsistencies in class semantics, annotation granularity, background definitions, and bounding box styles (Chen et al., 2023; Wang et al., 2019), as illustrated in Figure 1. These inconsistencies reduce downstream performance, especially when high accuracy on a specific target dataset is required (Liao et al., 2024). Manual relabelling is often infeasible at scale and, when label definitions diverge significantly, may be as costly as annotating from scratch.

**Limitations of Existing Approaches**  Model-centric solutions attempt to unify datasets via shared label spaces or embedding-based representations. Such methods rely on vision-language alignment (Radford et al., 2021; Ilharco et al., 2021; Shi et al., 2024; Chen et al., 2023; Zhou et al., 2023; Meng et al., 2023) or graph-based ontology construction (Xu et al., 2020; Ma et al., 2024) to harmonise labels. While effective for average-case generalisation, they do not prioritise fidelity to a specific target label space, limiting their utility in deployment scenarios that require strict semantic consistency.

Data-centric approaches instead project annotations from source datasets into a designated target label space (Liao et al., 2024; Lambert et al., 2020). However, they often rely on manual remapping (Lambert et al., 2020), support only one-to-one dataset transfers, or align bounding boxes without addressing semantic mismatches (Liao et al., 2024). Such limitations hinder scalability and applicability in real-world multi-dataset settings where both semantic (label definitions) and spatial (bounding box styles) inconsistencies are present.

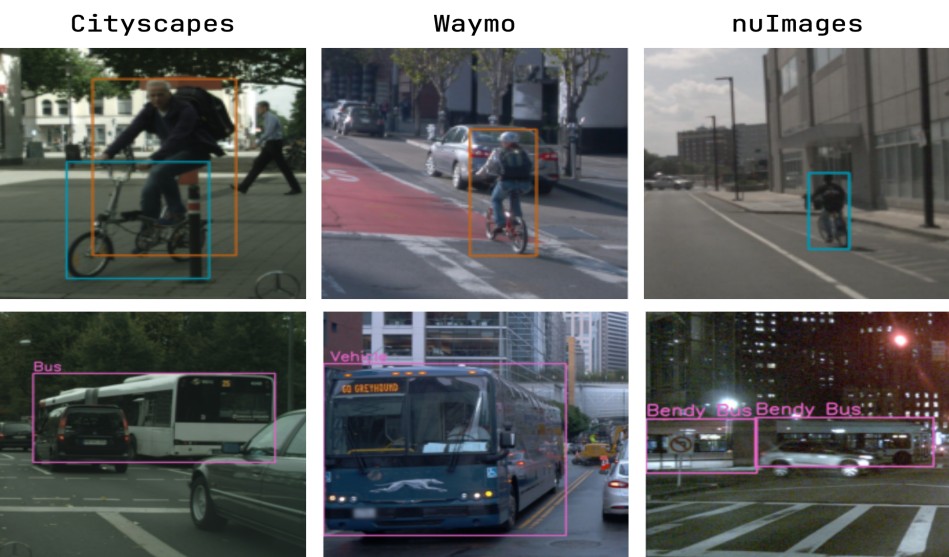

Figure 1: Annotation discrepancies between three road-based object detection datasets: Cityscapes, Waymo, and nuImages. The top row highlights differences in cyclist-related annotations: Waymo and nuImages treat the cyclist and bicycle as a single entity but assign different class labels, whereas Cityscapes annotates them separately. The bottom row illustrates differences in label granularity, with nuImages exhibiting the most fine-grained annotations and Waymo the least.

**Our Solution**   We propose a data-centric framework called **Label-Aligned Transfer (LAT)** for multi-dataset object detection. Rather than enforcing a unified label space, LAT projects annotations from multiple source datasets into the label space of a fixed target dataset. This enables practitioners to improve performance on a small, task-specific dataset by leveraging larger external datasets, without modifying model architectures or compromising the semantic integrity of the target label space. Crucially, LAT handles both class-level (semantic) and box-level (spatial) inconsistencies, allowing integration of heterogeneous datasets.

Our approach begins by training a separate detection model for each dataset, allowing specialisation in each dataset's native annotation style. These models generate cross-dataset pseudo-labels by predicting annotations in the label spaces of the other $N-1$ datasets, resulting in $N(N-1)$ pseudo-label projections. These projections act as implicit bridges between datasets. Using ground-truth annotations as supervision, we treat pseudo-labels as noisy alignment hints and learn semantic and spatial correspondences via multi-source training. This mitigates dataset-specific noise and leverages complementary supervision.

To enable effective transfer, LAT introduces two architectural modules: (1) a **Privileged Proposal Generator (PPG)** that replaces the traditional region proposal network by injecting fused proposals from both ground-truth and pseudo-labels, and (2) a **Semantic Feature Fusion (SFF)** module that refines region-level features using class-aware, overlap-sensitive attention to suppress noisy predictions and enhance feature alignment.

**Our Novelty**   Unlike conventional pseudo-labeling (Lee, 2013) or semi-supervised learning methods (Liu et al., 2021; Kennerley et al., 2023; 2024; Li et al., 2022; Hoffman et al., 2018), which often discard source-specific semantics, our approach preserves and aligns the information content of source labels to the target conventions. Rather than assuming shared label taxonomies or relying on manual mapping, LAT fuses pseudo-labels and ground-truth boxes to support direct semantic and spatial transfer.

Our PPG module enables joint proposal-level alignment, while SFF enhances feature consistency across datasets. In contrast to ontology-based (Xu et al., 2020; Ma et al., 2024) or embedding-based (Shi et al., 2024; Zhou et al., 2023; Radford et al., 2021) approaches, LAT avoids semantic drift and maintains dataset-specific granularity. To our knowledge, LAT is the first framework to

jointly resolve semantic and spatial inconsistencies in a multi-source, fixed-target detection setting without requiring unified taxonomies or manual relabeling.

In summary, our main contributions are:

- We propose **Label-Aligned Transfer (LAT)**, a novel data-centric framework that enables many-to-one label projection across heterogeneous detection datasets, without requiring shared taxonomies or manual remapping. Unlike existing methods, LAT jointly resolves class-level and spatial inconsistencies via integrated proposal and feature alignment.

- We design two novel architectural modules: the **Privileged Proposal Generator (PPG)** replaces the standard RPN by injecting both ground-truth and pseudo-labels, and the **Semantic Feature Fusion (SFF)** refines region features through class-aware attention over overlapping proposals. Together, they mitigate noise and enable cross-dataset supervision.

- LAT outperforms strong semi-supervised and label-transfer baselines across diverse class-divergent and scale-divergent settings, achieving up to +4.9 AP and +8.4 AP improvements. We further validate our architectural design through detailed ablation studies.

## 2 RELATED WORK

**Dataset Alignment** Integrating datasets for object detection presents challenges beyond visual domain shifts, including semantic misalignment and inconsistent annotation protocols. Early work included domain adaptation which focused on aligning image distributions via model-level adjustments such as Maximum Mean Discrepancy (MMD) (Yan et al., 2017), domain-adversarial training (Hoffman et al., 2018), and self-training (Kennerley et al., 2024; 2023; Li et al., 2022). Other approaches adopt a data-centric view, employing image translation to harmonise low-level appearance features across domains (Zheng et al., 2020; Chattopadhyay* et al., 2023). However, these methods primarily address distributional image-level variance and do not resolve inconsistencies in annotation semantics or structure. Annotation mismatches have been more extensively studied in classification (Recht et al., 2019; Beyer et al., 2020; Yun et al., 2021) and semantic segmentation (Bevandic and Segvic, 2022; Bevandić et al., 2022; Rottmann and Reese, 2023; Ma et al., 2024), while object detection remains underexplored (Liao et al., 2024). Our framework addresses this gap by jointly correcting semantic and spatial inconsistencies via direct label transfer into a designated target label space, without requiring manual taxonomies or repeated re-labelling.

**Multi-Dataset Object Detection** Multi-dataset training is commonly used to improve robustness and expand object category coverage (Chen et al., 2023; Ma et al., 2024; Meng et al., 2023). Approaches can be grouped into three broad categories: **(1)** partitioned detectors with dataset-specific heads (Zhou et al., 2022; Shi et al., 2024), **(2)** unified detectors trained on merged label spaces (Wang et al., 2019; Chen et al., 2023), and **(3)** hybrid models that incorporate pseudo-labelling across datasets (Liao et al., 2024). To unify label semantics, early work relied on manual class mapping and taxonomy construction (Lambert et al., 2020), while more recent approaches use vision-language models (Radford et al., 2021; Ilharco et al., 2021) to build shared embedding spaces, enabling prompt-based alignment across datasets (Shi et al., 2024; Chen et al., 2023; Zhou et al., 2023; Meng et al., 2023). While effective at harmonising class names, these methods often overlook differences in annotation coverage or bounding box conventions, and typically generate generalised label spaces rather than adapting to a task-specific target label space. In contrast, LAT transfers annotations directly into a fixed target label space, eliminating the need for dataset-specific heads or handcrafted taxonomies. Unlike previous methods that aim to optimise average performance across datasets, our approach is designed to maximise target-domain performance critical for real-world deployments with specific annotation requirements.

## 3 PROPOSED METHOD

We propose **Label-Aligned Transfer (LAT)** to address the challenges of merging object detection datasets with inconsistent label definitions. Our *data-centric framework* transfers annotations from multiple source datasets into the label space of a designated target dataset, without relying on a unified label taxonomy. This is achieved through a process of *collaborative pseudo-labeling*, where

each dataset-specific model generates predictions in other label spaces. These predictions, along with the ground-truth annotations, are used to learn correspondences at both the class and bounding box levels. In this way, LAT enables the transfer of both semantic and spatial information across datasets, effectively bridging inter-dataset discrepancies without requiring manual relabeling.

## 3.1 Preliminaries

We begin by formalizing the problem setup and outlining the procedure for generating initial pseudo-labels across datasets with divergent label spaces. To ensure clarity, we define the key terminology used throughout this paper:

- *Label Space:* The annotation style of a dataset, encompassing both semantic definitions (class labels) and spatial conventions (bounding boxes).
- *Ground-truth:* Human-annotated labels provided within a dataset, expressed in the dataset's native label space.
- *Upstream Model:* Object detection models trained independently on each dataset, used to generate pseudo-labels for images from other datasets.
- *Downstream Model:* The final object detection model trained using pseudo-labels that have been aligned to the target label space via LAT.

**Problem Formulation** Given $N$ datasets $\{D_1, D_2, \ldots, D_N\}$, with respective label spaces $\{L_1, L_2, \ldots, L_N\}$, our goal is to transfer annotations from all datasets into the label space of a target dataset, formalised as $L_{-N} \rightarrow L_N$. We assume that the datasets differ not only in bounding box conventions but also in class semantics. This substantially increases the difficulty of label transfer compared to prior approaches, which assume a shared class labels across datasets (Liao et al., 2024).

Our label transfer framework operates on a triplet $\{I_n, PL_{-n}^n, GT_n\}$, where, $I_n$ represents the images from the $n$-th dataset; $PL_{-n}^n$ represents pseudo-labels of the images in the label space of all other datasets, $L_{-n}$; and $GT_n$ is the corresponding ground-truth labels of dataset $n$. After training, LAT outputs a refined set of pseudo-labels aligned to a designated target label space. Ultimately, our objective is to generate target-aligned annotations for all datasets, enabling downstream training of object detectors within a consistent label space.

The primary challenge lies in the absence of paired supervision: ground-truth annotations are never observed in both source and target label spaces for the same image. This precludes any direct mapping between class definitions or bounding box conventions. In addition, datasets may exhibit class label sparsity, semantic overlaps, and inconsistent naming conventions. To address these challenges, LAT performs many-to-one label space transfer, allowing pseudo-labels from different datasets to reinforce each other in a collaborative, ensemble-like training process. This collaborative supervision also helps mitigate the noise inherent in individual pseudo-labels.

**Generating Initial Pseudo-labels** For each dataset $D_n$, we train a corresponding upstream detection model $M_n$, optimized on its native label space $L_n$. Using these trained upstream models, we generate pseudo-labels for each dataset under every other dataset's label space. As a result, each dataset yields $N-1$ sets of pseudo-labels, corresponding to the annotation formats of the remaining datasets, denoted by $L_{-n}$. These pseudo-labels are accompanied by classification confidence scores, which are retained for upstream model training. To enhance reliability, we apply non-maximum suppression and score thresholding to discard low-confidence predictions.

These initial pseudo-labels serve as cross-space predictions for each dataset. However, they are generated independently by each upstream model and do not benefit from additional contextual signals that could improve their quality. We refer to such signals as *privileged information*, which is typically unavailable in standard supervised training. In our framework, privileged information provided to LAT includes ground-truth annotations (classes and bounding boxes) of the current images as well as pseudo-labels from other label spaces.

## 3.2 Label-Aligned Transfer

Our Label-Aligned Transfer (LAT) framework, Figure 2, extends a standard two-stage object detector to incorporate privileged information during both training and inference. A typical two-stage detector

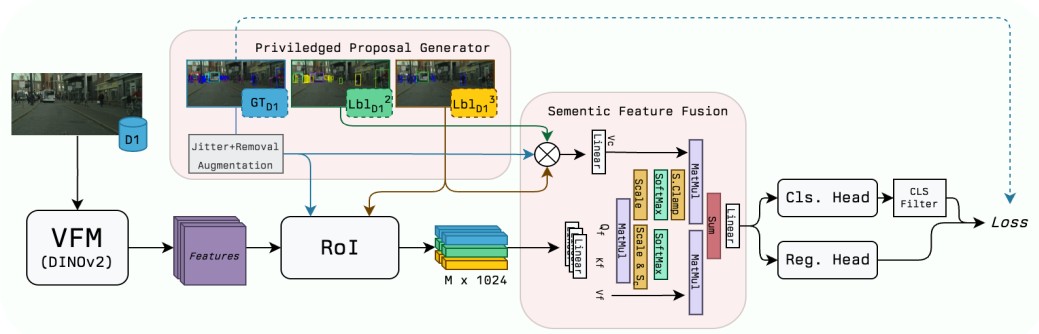

Figure 2: Overview of the LAT architecture. Dataset-specific pseudo-labels and ground-truth annotations are combined via the Privileged Proposal Generator (PPG), which replaces the region proposal network. A frozen Vision Foundation Model (VFM) extracts shared image features. The Semantic Feature Fusion (SFF) module then refines region features by injecting class-aware information using attention over overlapping proposals. We filter the classification output to compute loss on only the current datasets label space.

(Ren et al., 2015) consists of three main components: a feature extractor $f_{img}$, a region proposal network (RPN), and a region-of-interest (RoI) pooling layer. The RPN generates class-agnostic bounding box proposals from the feature maps, while the RoI layer extracts fixed-size region features that are passed to the classification and regression heads.

In LAT, we replace the conventional feature extractor with a frozen Vision Foundation Model (VFM), such as DINOv2 (Oquab et al., 2024). We further substitute the RPN with our **Privileged Proposal Generator (PPG)**, which provides *privileged information*-based proposals derived from both ground-truth and pseudo-label sources. These proposals are passed to the RoI layer and also serve as input to the **Semantic Feature Fusion (SFF)** module, which refines region features using class-aware attention.

### 3.2.1 PRIVILEGED PROPOSAL GENERATOR (PPG)

Our Privileged Proposal Generator (PPG) replaces the outputs of a RPN with pseudo-labels generated as described in Section 3.1, alongside the ground-truth labels for each image in the training batch. We apply light augmentations to the ground-truth labels, such as random jittering and selective removal of bounding boxes. These augmented labels are then used by the RoI layer to crop region features from the shared feature map. Since these labels are derived from multiple label spaces, they often contain overlapping objects across datasets, regardless of naming convention. For example, the concept of a *car* may appear across datasets but be labeled as *vehicle* in Waymo and *car* in Cityscapes. Such overlaps provide a rich supervisory signal and are critical to the effectiveness of our Semantic Feature Fusion (SFF) module.

In addition to supplying bounding box proposals to the RoI layer, PPG also outputs the associated class labels for each region to the SFF module. To maintain label discreteness, we concatenate the label sets from all datasets instead of merging classes with identical names. This ensures that semantically divergent classes, despite having the same label name, are not erroneously unified.

By leveraging both ground-truth and cross-space pseudo-labels, PPG exposes LAT to diverse annotation styles. In turn, this provides essential information for learning semantic and spatial correspondences across label spaces.

### 3.2.2 SEMANTIC FEATURE FUSION (SFF)

To improve cross-dataset feature consistency, we introduce the Semantic Feature Fusion (SFF) module (Figure 3). SFF enhances region features by attending over overlapping proposals and injecting class-aware information from both pseudo-labels and ground-truth annotations. This allows the model to learn semantic relationships between related but differently labelled classes.

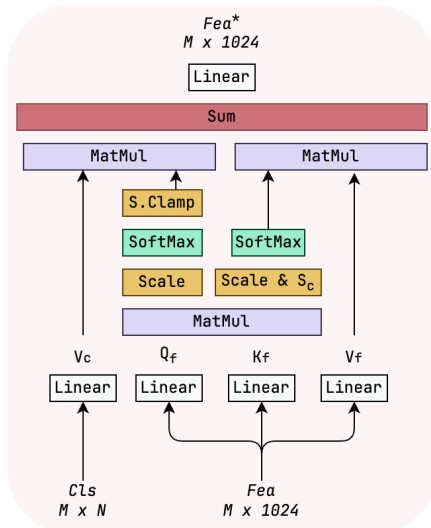

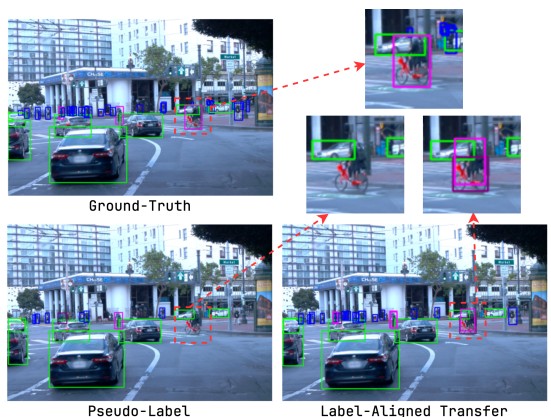

Figure 4: LAT addresses noisy pseudo-labels by leveraging ground-truth supervision and multi-source consensus. Ground-truth labels (Waymo) are shown alongside pseudo-labels and LAT predictions (Cityscapes label space). LAT correctly separates the cyclist and bicycle into distinct classes and boxes, resolving semantic conflation. Ground-truth anchoring also enables recovery of small objects missed by pseudo-labels.

Figure 3: Our Semantic Feature Fusion (SFF) module leverages an attention mechanism to fuse and inject information from overlapping proposals. This enhances the capacity of the classification and regression heads to model relationships between class labels from distinct label spaces.

Let $A \in \mathbb{R}^{M \times M}$ denote the attention matrix computed using scaled dot-product attention over RoI features:

$$A = \frac{QK^T}{\sqrt{d}},$$

where $Q, K \in \mathbb{R}^{M \times d}$ are learned projections of the RoI features. Let $V_c \in \mathbb{R}^{M \times d}$ be the value matrix derived from classification scores after linear projection, and $V_r \in \mathbb{R}^{M \times d}$ be the value matrix derived from region features after linear projection. We define a confidence vector $S_c \in \mathbb{R}^M$, where each entry is set to 1 for ground-truth proposals, or $\max(C_m)$ for pseudo-label proposals, where $C_m$ is the classification score vector of the $m$-th proposal. $S_c$ is applied to the attention matrix of the feature branch to prioritise more confident pseudo-labels.

To mitigate noisy pseudo-labels across datasets, we apply a row-wise scaling mechanism to the attention matrix used in the classification branch. Specifically, each row of the attention matrix $A$ is scaled such that its maximum value is capped at a threshold $T = 1/\sqrt{N}$, where $N$ is the number of datasets. This promotes the aggregation of information from overlapping pseudo-labels while suppressing those that occur independently and are more likely to be erroneous.

The final fused feature representation is computed as:

$$SA = \text{clamp}\left(\text{softmax}(A)\right) V_c + \text{softmax}(S_c \circ A) V_r,$$

where $\circ$ denotes element-wise multiplication. Softmax is applied row-wise, and clamping ensures that each row's maximum value does not exceed $T$. This fused output provides the downstream classifier with enriched semantic and visual features, improving cross-dataset generalisation.

SFF enables the classification and regression heads of LAT to leverage enriched visual features and semantic context from overlapping label spaces. During training, we mask the classification logits, prior to loss calculation, to include only the classes present in the current batch, ensuring that intra-dataset supervision remains dominant while still benefiting from inter-dataset relationships. At inference, logits are masked to the designated target label space. By learning cross-dataset semantic correspondences, LAT can generate robust predictions even for datasets not originally labelled under the target space. We demonstrate the effectiveness of this approach in Section 4.

Due to the collaborative nature of SFF, our method is able to mitigate the impact of noisy pseudo-labels. SFF learns to identify and reinforce consistent object predictions when multiple pseudo-labels corroborate the presence of an object shared across label spaces. This alignment is further anchored by the ground-truth annotations when the object is also labelled in the target dataset. An example of pseudo-label refinement is shown in Figure 4, with additional results included in the supplementary materials.

## 4 EXPERIMENTS

We conduct extensive experiments to evaluate the effectiveness of our proposed Label-Aligned Transfer (LAT) framework in realistic multi-dataset object detection scenarios. Our evaluations focus on two primary challenges: (1) semantic inconsistencies due to divergent class taxonomies across datasets, and (2) performance degradation in small datasets when augmented with larger ones. These benchmarks simulate practical conditions where direct dataset merging would be ineffective or harmful.

### 4.1 BENCHMARKS

**Cityscapes ↔ nuImages ↔ Waymo.** This benchmark targets the challenge of **class divergence** across datasets. Cityscapes (Cordts et al., 2016), nuImages (Caesar et al., 2019), and Waymo (Sun et al., 2020) contain 8, 24, and 3 annotated classes, respectively, with varying levels of granularity. To isolate the effects of class variability, we subsample 3,000 images from each dataset.

**Cityscapes ↔ ACDC ↔ BDD100K ↔ SHIFT.** This benchmark evaluates performance under a **small-versus-large dataset** setting Cityscapes (Cordts et al., 2016) and ACDC (Sakaridis et al., 2021) represent small-scale datasets, each comprising of *2,965* and *1,571* samples, respectively. In contrast, BDD100K (Yu et al., 2020) and SHIFT (Sun et al., 2022) are large-scale datasets, each containing *69,852* and *141,052* images.

Further details of benchmark selection and datasets are available in the supplementary materials.

### 4.2 EXPERIMENTAL SET-UP

We implement LAT using the FRCNN (Ren et al., 2015) framework built on Detectron2 (Wu et al., 2019). DINOv2 (Oquab et al., 2024) is employed as a frozen feature extractor with pre-trained weights. For downstream training, we employ FRCNN and RT-DETR (Zhao et al., 2024) detection models. All downstream models are trained using four NVIDIA RTX 3090 GPUs. Further training details can be found in the supplementary materials.

**Baselines.** We compare LAT against several methods, including a baseline without label transfer, two semi-supervised approaches: student-teacher supervision and pseudo-labelling, and a label unification approach.

- *Baseline:* A model trained solely on the target dataset using its native label space.
- *Student-Teacher* (Liu et al., 2021): Trained with full supervision on the target dataset and semi-supervised loss on unlabelled samples from other datasets.
- *Pseudo-Label* (Lee, 2013): A model is first trained on the target dataset and then used to generate pseudo-labels on other datasets. These labels are filtered using non-maximum suppression and confidence thresholding before being used in continued training.
- *Plain-DET* (Shi et al., 2024): A label unification approach build on the Deformable DETR (Zhu et al., 2020) framework for object detection datasets.

All baseline models adopt exponential moving average (EMA) updates for fair comparison.

### 4.3 RESULTS

**LAT improves performance across benchmark scenarios.** As shown in Table 1, LAT significantly outperforms the baseline in scenarios involving datasets with differing class granularity. It also yields

Table 1: Performance on the Class Divergence Benchmark. Downstream detector trained on LATs pseudo-labels outperforms all baselines (FRCNN and RT-DETR) in datasets with differing class granularities. Cityscapes, despite moderate class count, sees the largest gain (+4.9 AP).

| | | | DATASET | | |
|---|---|---|---|---|---|
| METHOD | MODEL | METHOD | Cityscapes | nuImages | Waymo |
| Baseline | FRCNN | - | 55.2 | 39.2 | 44.6 |
| Student-Teacher | FRCNN | Semi-Supervised | 55.1 | 40.1 | 44.2 |
| Pseudo-Label | FRCNN | Label Transfer | 56.9 | 40.6 | 45.6 |
| LAT | FRCNN | Label Transfer | **60.1** | **41.7** | **48.5** |
| Baseline | RT-DETR | - | 56.8 | 37.0 | 45.3 |
| Plain-DET | Def-DETR | Label Unification | 52.2 | 22.0 | 43.6 |
| LAT | RT-DETR | Label Transfer | **60.6** | **39.5** | **49.6** |

Table 2: Performance on the Small-to-Large Dataset Benchmark. LAT improves small datasets and scales with training. LAT shows large gains on ACDC (+8.4 AP), while large datasets experience minor drops without longer training. Longer training recovers performance.

| | | | DATASET | | | |
|---|---|---|---|---|---|---|
| METHOD | MODEL | METHOD | Cityscapes | ACDC | BDD100K | SHIFT |
| Baseline | FRCNN | - | 55.2 | 45.0 | 57.2 | 69.9 |
| Student-Teacher | FRCNN | Semi-Supervised | 55.4 | 48.2 | 56.2 | 68.6 |
| Pseudo-Label | FRCNN | Label Transfer | 58.5 | 50.7 | 56.1 | 68.9 |
| LAT | FRCNN | Label Transfer | 60.0 | **53.4** | 56.1 | 69.3 |
| LAT (Long Train) | FRCNN | Label Transfer | **60.2** | 53.3 | **57.8** | **71.4** |
| Baseline | RT-DETR | - | 56.8 | 43.6 | 57.3 | 69.5 |
| Plain-DET | Def-DETR | Label Unification | 55.5 | 49.0 | 50.5 | 58.5 |
| LAT | RT-DETR | Label Transfer | **61.2** | **49.0** | 53.1 | 65.7 |
| LAT (Long Train) | RT-DETR | Label Transfer | 59.8 | 47.9 | **58.1** | **69.9** |

substantial gains for smaller datasets when combined with much larger ones, as illustrated in Table 2. While we observe a slight performance drop for large datasets, which may be attributed to under fitting, this is easily address by increasing the training iterations.

**Domain gap limits student-teacher performance.** When compared to other models that use exponential moving average (EMA) without full semi-supervised training, student-teacher approaches consistently underperform, even falling below the baseline. One likely explanation is that domain gaps between datasets cause pseudo-label errors to propagate through the teacher model during training (Li et al., 2022; Kennerley et al., 2023). In contrast, LAT avoids this issue by retaining ground-truth labels during label transfer, which serve as reliable anchors for supervising pseudo-label refinement.

**Label Transfer improves robustness to noisy supervision.** We observe that detectors trained on LAT-generated labels consistently outperform those trained on standard pseudo-labels. This highlighting LAT's ability to mitigate noise introduced during pseudo-label generation. This robustness stems from LAT's integration of ground-truth annotations and multi-source pseudo-labels, allowing the model to resolve both semantic and spatial inconsistencies. In our supplementary material, we provide qualitative examples where LAT corrects various forms of pseudo-label noise, including missing annotations, misclassified or misaligned boxes, and false positives.

**Fine-tuning remains most effective.** We evaluate several strategies for training the downstream detector in Table 3. These strategies vary the composition of source and target datasets within training batches: *50/50 Batch* refers to batches containing an equal number of samples from each domain, while *Mixed Batch* denotes random mixing of source and target samples. *Fine-Tuning* modifies the

Table 3: Training strategy comparison for down-stream detectors. Fine-tuning after mixed pre-training yields the highest gains, confirming the value of domain-specialized adaptation.

| Training | DATASET | | |
|---|---|---|---|
| | Cityscapes | nuImages | Waymo |
| 50/50 Batch | 57.5 | 40.5 | 46.8 |
| Mixed Batch | 58.9 | 40.0 | 47.1 |
| Fine-Tuning | **60.1** | **41.7** | **48.5** |

Table 4: Ablation of attention mechanism in SFF. SFF with class-aware attention clearly outperforms no attention or standard attention across all datasets.

| | DATASET | | |
|---|---|---|---|
| | Cityscapes | nuImages | Waymo |
| No Attn. | 57.5 | 39.8 | 46.1 |
| Standard Attn. | 58.1 | 40.4 | 47.0 |
| SFF | **60.1** | **41.7** | **48.5** |

Mixed Batch regime by replacing the final 10,000 training iterations with batches containing only source dataset samples. While all strategies outperform the baseline, we observe that pretraining on a mixed dataset followed by fine-tuning leads to the most consistent gains, highlighting the benefit of learning generaliable features before domain-specific adaptation.

**Injecting class information improves performance.** Table 4 compares three variants of the attention mechanism used within LAT: no attention, standard attention, and our proposed Semantic Feature Fusion (SFF), which incorporates class-aware weighting. We observe consistent gains across all datasets when class information is injected into the attention computation. This highlights the value of injecting class-level context during feature fusion, particularly in scenarios where semantically similar objects, such as 'car' and 'vehicle', are annotated under distinct labels across datasets, despite representing the same underlying object.

Table 5: Scaling attention in SFF: Class vs. Feature. Applying scaling to the class branch gives better performance than feature branch or no scaling.

| | DATASET | | |
|---|---|---|---|
| | Cityscapes | nuImages | Waymo |
| No Scaling | 57.8 | 40.9 | 47.3 |
| Feature Scaling | 59.6 | 41.5 | 47.9 |
| Class Scaling | **60.1** | **41.7** | **48.5** |

**Class scaling outperforms feature scaling.** In Table 5, we compare different scaling strategies for the softmax attention weights in our Semantic Feature Fusion module. Specifically, we evaluate whether scaling is more effective when applied to the feature branch or the class branch. While feature scaling yields improvements over no scaling, applying the scaling to the class branch consistently performs best. One possible explanation is that the feature branch already incorporates confidence-based weighting through the $S_c$ term, making additional scaling less beneficial.

Additional experiments are presented in the supplementary materials.

## 5 CONCLUSION

Label-Aligned Transfer (LAT) addresses the challenge of integrating object detection datasets with heterogeneous label spaces. LAT modifies the standard two-stage detector by replacing the region proposal network with a *Privileged Proposal Generator* (PPG), which incorporates both ground-truth and pseudo-label proposals from multiple source label spaces. A *Semantic Feature Fusion* (SFF) module further refines region features by injecting privileged, class-aware context via attention over overlapping proposals. Our experiments demonstrate that LAT consistently improves target-domain performance, with gains of +4.9AP and +8.4AP on benchmarks evaluating class-divergent and scale-divergent dataset transfer settings, respectively. In addition to achieving state-of-the-art results in complex multi-dataset scenarios, LAT effectively mitigates pseudo-label noise and generalises well across diverse detector architectures.

**Limitations.** Our method assumes annotations are required solely in a fixed target label space. Merging multiple label spaces or extending the target ontology is not yet supported, though future work may explore more flexible transfer. While LAT introduces no inference-time cost, scaling to many datasets incurs pre-processing overhead during initial pseudo-label generation.

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

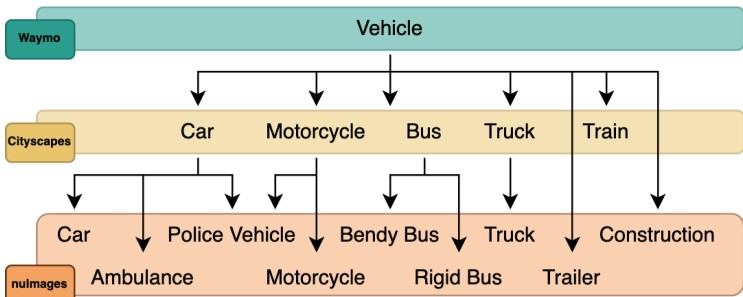

Figure 5: Illustration of class granularity divergence for vehicle-related labels across Waymo, Cityscapes, and nuImages. Waymo uses broad classes, while nuImages is the most fine-grained.

# A  APPENDIX

## A.1  SCALING LABEL-ALIGNED TRANSFER

LAT adopts a modular multi-stage pipeline designed for flexible dataset integration. In the first stage, an object detector is trained independently for each of the N datasets, scaling linearly with N and enabling parallel training. In the second stage, pseudo-labels are generated by running each model on the remaining N-1 datasets, resulting in N(N-1) inference passes. While this introduces quadratic scaling in pre-processing, the computation is parallel and easily distributed.

This design provides two key benefits: (1) it preserves dataset-specific annotation conventions before unification. (2) it allows components to be updated independently without retraining the entire system. We view this overhead as a strategic trade-off for high-fidelity label transfer, particularly suited for scenarios involving an intermediate number of datasets. While scaling beyond this range may pose challenges, these are confined to the training phase. A key advantage of LAT is that it introduces no additional overhead at inference time, once trained, the downstream detector operates with standard runtime efficiency.

## A.2  ADDITIONAL EXPERIMENTAL DETAILS & EXPERIMENTS

### A.2.1  ADDITIONAL DETAILS ON BENCHMARKS

**Cityscapes ↔ nuImages ↔ Waymo.**   This benchmark targets the challenge of label granularity divergence across datasets, i.e., when datasets refer to the same object category using semantically or structurally distinct label taxonomies. Cityscapes (Cordts et al., 2016), nuImages (Caesar et al., 2019), and Waymo (Sun et al., 2020) offer a compelling testbed due to their highly mismatched annotation conventions, containing 8, 24, and 3 annotated classes, respectively. Waymo provides broad semantic categories like *vehicle*, which subsumes finer-grained classes from other datasets. For instance, its *vehicle* label covers five distinct Cityscapes classes (e.g., *car*, *bus*, *truck*) and nine from nuImages (e.g., *trailer*, *construction vehicle*). In contrast, nuImages is highly discrete in its class definitions—even annotating compound labels like *police vehicle*, which would be split into *car* and *motorcycle* in Cityscapes. Figure 5 illustrates an example of vehicle super-class. These semantic discrepancies are further exacerbated by visual overlap in the object regions, making alignment particularly challenging. To ensure controlled evaluation, we subsample 3,000 images from each dataset and hold the label mapping fixed throughout all runs. This allows us to isolate the effects of semantic transfer without confounding factors from sample size.

**Cityscapes ↔ ACDC ↔ BDD100K ↔ SHIFT.**   This benchmark investigates the impact of dataset size disparity, a frequent scenario in practice, where a small, curated target dataset is augmented with large-scale external sources. Cityscapes (Cordts et al., 2016) and ACDC (Sakaridis et al., 2021) represent small but high-quality datasets with strong label fidelity, comprising *2,965* and *1,571* images, respectively. In contrast, BDD100K (Yu et al., 2020) and SHIFT (Sun et al., 2022) serve as large-scale sources with *69,852* and *141,052* images, respectively. SHIFT is a synthetic dataset, providing domain diversity without additional labelling cost, a common practice in modern

data-centric pipelines. Importantly, all four datasets share consistent class definitions, making this benchmark well-suited to isolate the impact of spatial annotation differences and sample imbalance, rather than semantic divergence.

From a deployment perspective, it is common to begin with a small, task-specific dataset that reflects the intended deployment domain, e.g., Cityscapes for urban driving or ACDC for adverse conditions, and then enhance performance by incorporating publicly available data such as BDD100K or SHIFT. Thus, the performance of Cityscapes and ACDC under label transfer is particularly relevant, as it simulates the realistic case of training a downstream model with limited annotations without modifying the target label space. This benchmark allows us to assess how well LAT supports such transfer without sacrificing annotation fidelity or overfitting to larger source domains.

### A.2.2 EXPERIMENTAL SET-UP

We implement LAT using the FRCNN (Ren et al., 2015) framework built on Detectron2 (Wu et al., 2019). DINOv2 (Oquab et al., 2024) is employed as a frozen feature extractor with pre-trained weights. In our PPG module, random jittering and ground-truth label removal are applied at rates of 0.5 and 0.05, respectively. LAT is trained for 30,000 iterations using a learning rate of 0.2 and a batch size of 4 on a single RTX 3090 GPU.

For downstream training, we use a FRCNN model with a modified weighted cross-entropy loss, where the weight is derived from the confidence score of the pseudo-label. This model, as well as the initial pseudo-label generation model, is trained for 50,000 iterations with a fixed learning rate of 0.2 and a batch size of 16. In addition, we train RT-DETR and YOLOv11 models as our downstream detector for comparisons to more modern detectors as compared to FRCNN. These detectors are trained for 300,000 iterations with a batch size of 64. AdamW is used as the optimizer with a learning rate of 0.001 and momentum of 0.9. All downstream models are trained using four NVIDIA RTX 3090 GPUs.

### A.2.3 RESULTS

Table 6: YOLO-based downstream detectors in class-divergent setting. LAT consistently improves performance even with a single-stage detector like YOLO.

| METHOD | MODEL | METHOD | DATASET | | |
| --- | --- | --- | --- | --- | --- |
| | | | Cityscapes | nuImages | Waymo |
| Baseline | YOLO | - | 53.9 | 37.1 | 45.6 |
| LAT | YOLO | Label Transfer | **59.1** | **37.6** | **47.2** |

Table 7: YOLO-based downstream detectors in small–large dataset setting. ACDC sees large gains under LAT, replicating trends seen with FRCNN and RT-DETR.

| METHOD | MODEL | METHOD | DATASET | | | |
| --- | --- | --- | --- | --- | --- | --- |
| | | | Cityscapes | ACDC | BDD100K | SHIFT |
| Baseline | YOLO | - | 53.9 | 41.4 | 56.0 | **64.2** |
| LAT | YOLO | Label Transfer | **57.9** | **42.1** | 50.1 | 60.1 |
| LAT (Long Train) | YOLO | Label Transfer | 58.2 | 43.7 | **56.4** | **64.2** |

**Consistent performance on YOLOv11.** We conduct additional experiments using YOLOv11 (Jocher et al., 2023) to verify that the performance gains from LAT's refined pseudo-labels are not tied to a specific model architecture. As shown in Table 6 and Figure 7, LAT-trained labels consistently improve performance on YOLOv11, complementing the results already demonstrated with FRCNN and RT-DETR in the main paper. This confirms that the benefits of LAT generalise across detectors, demonstrating its effectiveness as a model-agnostic label transfer framework.

Table 8: Performance on Synscapes → Cityscapes (1:1 transfer). LAT matches the performance of state-of-the-art LGPL despite the simpler one-to-one transfer setup.

| Model | Def-DETR | Faster RCNN |
|---|---|---|
| Baseline | 32.9 | 38.7 |
| Pseudo-Label | 30.7 | 36.9 |
| Pseudo-Label + Filtering | 33.0 | 39.1 |
| LGPL (Liao et al., 2024) | 34.5 | **39.7** |
| LAT | **34.7** | 39.6 |

**Performance on simpler transfer protocols.** We compare our method to LGPL (Liao et al., 2024) in Table 8, using the synthetic-to-real transfer setting from Synscapes (Wrenninge and Unger, 2018) to Cityscapes (Cordts et al., 2016). We consider this a simpler transfer scenario, as both datasets share identical class labels and exhibit similar semantic structures. Moreover, the setup involves a one-to-one label space transfer, reducing the need for LAT's full design capabilities, such as many-to-one label alignment and the performance gains that emerge from integrating multiple source datasets. Nevertheless, LAT matches the performance of the state-of-the-art LGPL method, demonstrating its effectiveness even under minimal transfer complexity. We note that LGPL results are reported directly from the original paper using mAP@[.5:.95] as a metric, as public code was not available at the time of writing.

**Qualitative results: LAT mitigates pseudo-label noise.** We illustrate LAT's effectiveness in addressing pseudo-label noise in Figures 6,7, and8. Each figure presents three columns: the first shows initial pseudo-labels from the upstream detector in the target label space; the second shows LAT-refined pseudo-labels in the same label space; and the third displays the ground-truth annotations in the original source label space. Note that class names may differ between the pseudo-labels and ground-truth columns due to label space discrepancies.

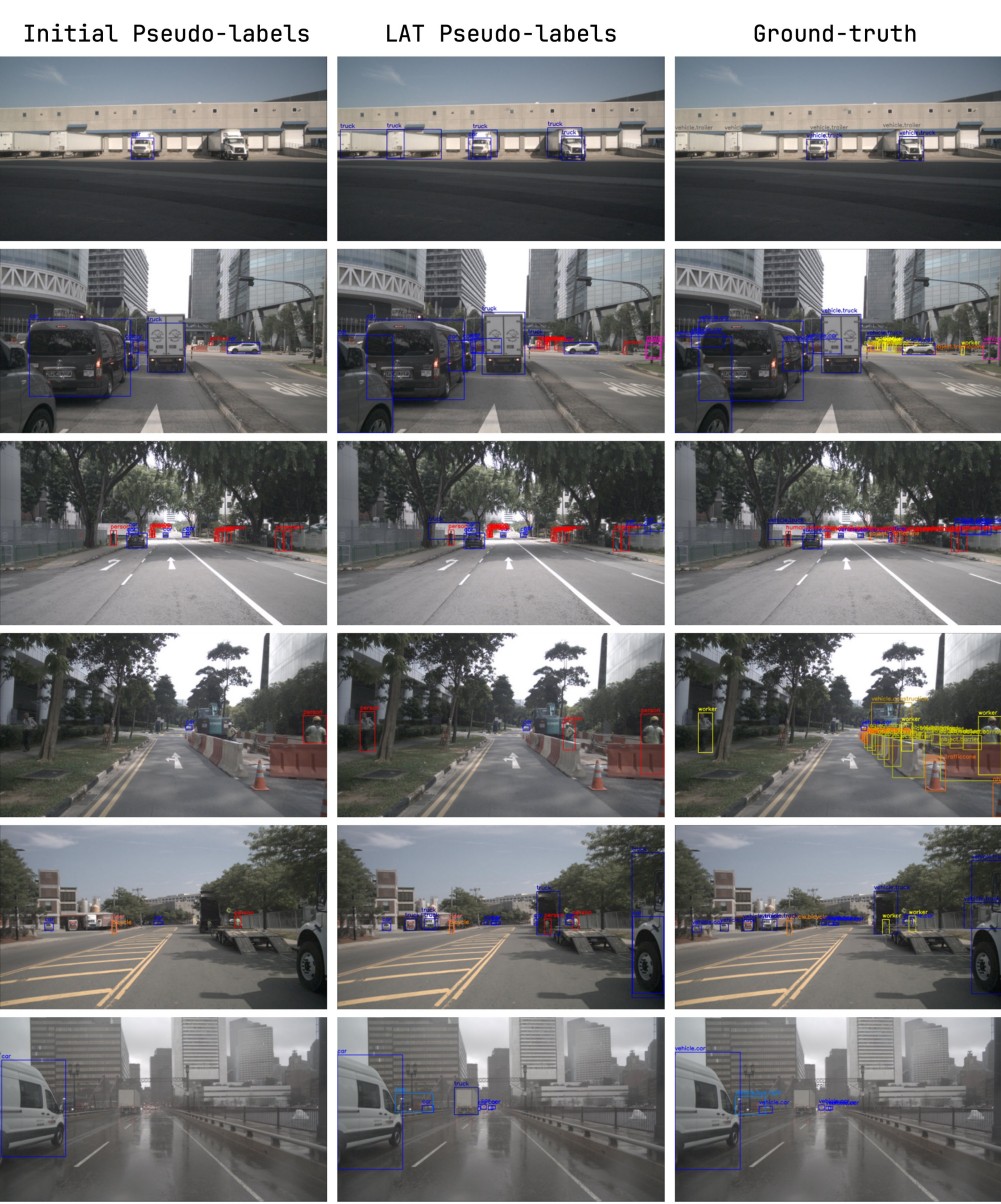

Figure 6: Qualitative results of Cityscapes target label space in class-divergent setting with nuImages dataset. **Row 1:** LAT recovers truck instances missing from initial pseudo-labels. **Row 2:** LAT refines truck bounding boxes and detects small objects. **Row 3:** LAT recovers heavily obscured cars. **Row 4:** LAT detects pedestrians omitted by upstream pseudo-labels. **Row 5:** LAT identifies foreground and background trucks. **Row 6:** LAT recovers vehicles under adverse weather conditions (rain).

Initial Pseudo-labels      LAT Pseudo-labels      Ground-truth

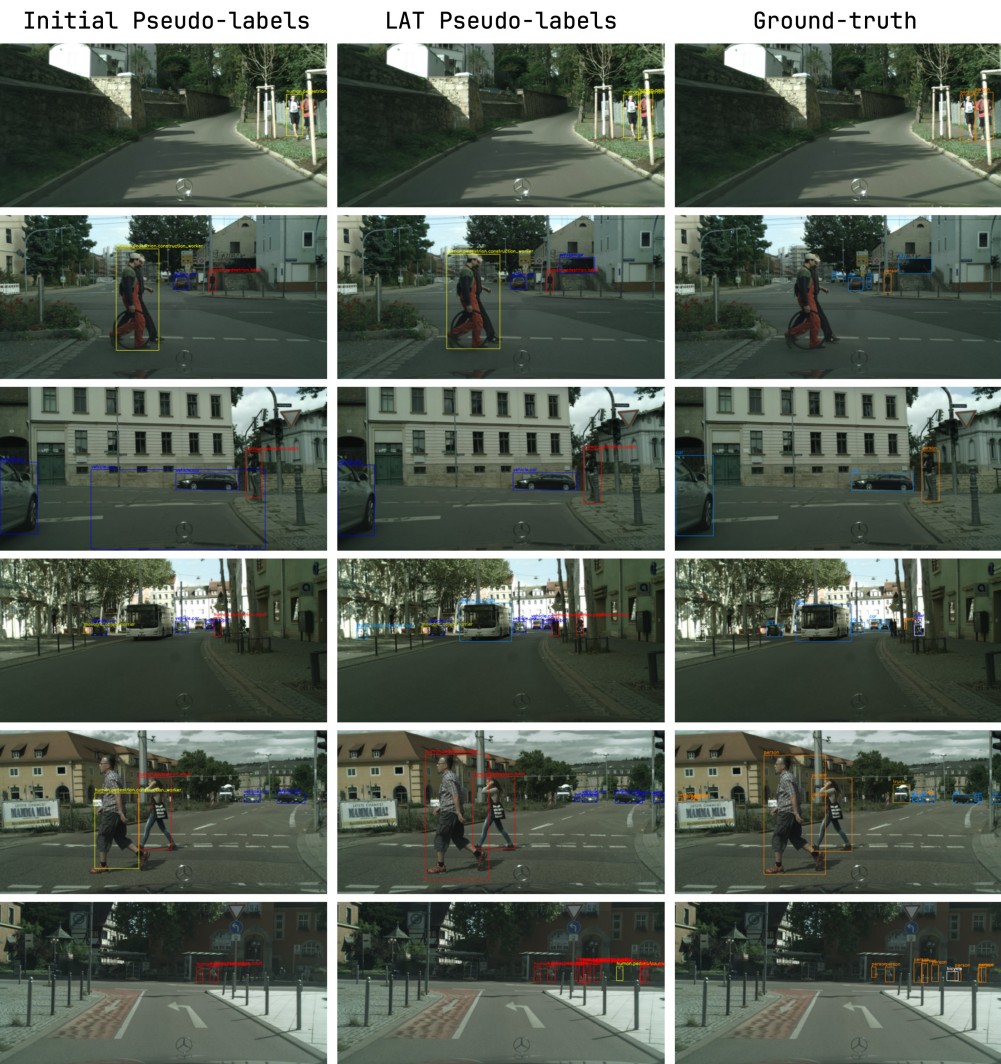

Figure 7: Qualitative results of nuImages target label space in class-divergent setting with Cityscapes dataset. **Row 1:** LAT detects an occluded pedestrian missed by initial pseudo-labels. **Row 2:** LAT corrects noisy human-annotated ground-truth. **Row 3:** LAT removes an erroneously predicted car from initial pseudo-labels. **Row 4:** LAT recovers a bus instance omitted by the upstream model. **Row 5:** LAT refines pedestrian bounding box and corrects its class label. **Row 6:** LAT recovers small-scale pedestrian and bicycle instances.

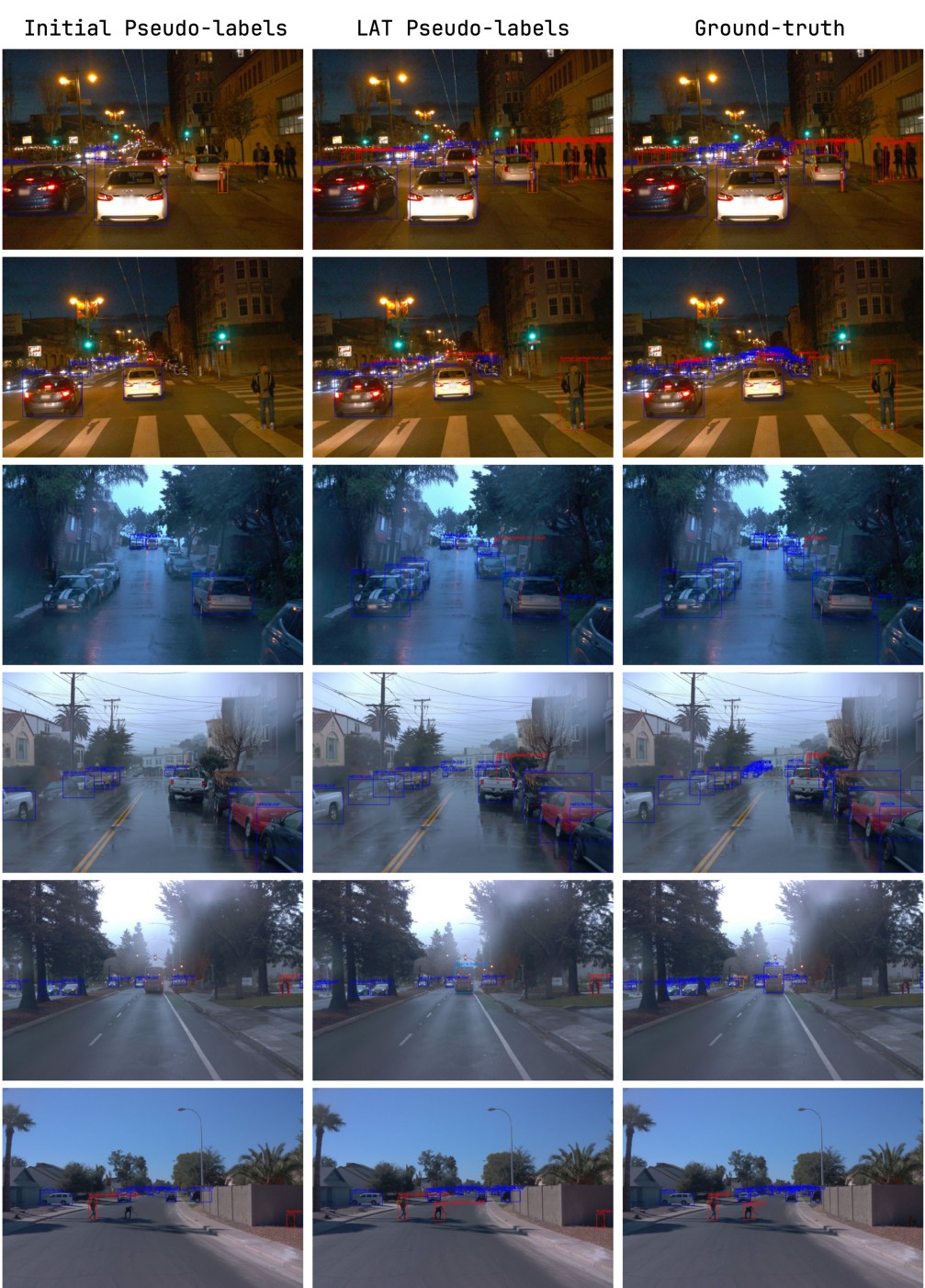

Figure 8: Qualitative results of nuImages target label space in class-divergent setting with Waymo dataset. **Row 1:** LAT detects background pedestrians and vehicles under adverse nighttime conditions. **Row 2:** LAT detects both foreground pedestrians and background vehicles at night. **Row 3:** LAT recovers multiple vehicles in rainy nighttime scenes. **Row 4:** LAT detects vehicles and pedestrians in rainy conditions. **Row 5:** LAT recovers multiple vehicle instances in rain. **Row 6:** LAT detects a pedestrian in an uncommon pose.

