# OpenReview forum: "Mind the Gap: Transferring Labels Across Datasets with Divergent Annotation Protocols"
_ICLR.cc/2026/Conference — ICLR 2026 Conference Withdrawn Submission_

### Official Review · Reviewer_aLf5 · 2025-10-17

**Soundness:** 2
**Presentation:** 1
**Contribution:** 2
**Rating:** 2
**Confidence:** 4

**Summary:**

The paper addresses annotation transfer for object detection in the autonomous driving domain, a task that involves adapting annotations from source datasets to a target one. The authors propose PPG, which is a RPN trained on both ground-truth and pseudo-labels. Their framework also incorporates a module named SFF, which leverages the attention mechanism to fuse features between ROI. The authors report that their experiments demonstrate the effectiveness of proposed methods.

**Strengths:**

The task of transferring annotations from existing large datasets to specific real-world scenarios is interesting and also meaningful in a practical sense

**Weaknesses:**

**Title**
- The paper title sets a broader expectation than the paper's specific focus. It is titled “transferring labels across datasets with divergent annotation protocols”, suggesting a general solution, however, the work itself focuses specifically on object detection for autonomous driving. A more specific title would be more appropriate.

**Presentation**

The paper's writing and presentation require significant improvement, as the current version is difficult to follow.

- One primary concern is the mathematical formalization and notation, which is often unclear and non-standard.
For example, several notational choices hinder readability:
  - The formalization of the problem as $L_{−N} \rightarrow L_N$ is not clear. The notation $L_{−N}$, presumably used to represent a set excluding elements related to N, is non-standard. Its meaning must be inferred from the context, which is not ideal. An explicit definition of this notation is crucial for clarity.
  - The notation $PL^n_{-n}$ appears to represent a single, simple object. However, based on the context, it seems to be a structured set associated with multiple pseudo-labels from different label spaces. This complexity should be clarified, and the notation should be revised to better reflect its structure.
  - $I_n$ and $GT_n$ should be explicitly denoted as sets, as they represent sets of images and ground truths for dataset $n$, respectively.
- Line 201: The paper states that an upstream model can generate pseudo-labels in the label spaces of other datasets. This is unclear to me considering a model optimized on a specific label space ($L_N$) should only be able to predict labels within that same space.
- Line 257-260: The text here describes the PPG outputs class labels, which is confusing, as PPG is introduced as a region proposal module. Standard RPNs are class-agnostic and only generate objectness scores and bounding box coordinates, not class-specific labels. While I understand the authors may be describing how class labels for the final head are formulated, attributing this function directly to an RPN-like module is misleading.
- Line 296: The dimension M of the attention matrix A is not defined , which is important for reader to understand the inputs and mechanics of the specific attention module.

**Experiments**
- The experiment section lacks detailed explanations of the setup. For example, regarding Table 1, it is unclear what training data was used when evaluating on nuImages. The authors must provide these details for each experiment to clarify what is being measured and to ensure the results are interpretable.
- As shown in Table 1, the mean-teacher based semi-supervised approach surprisingly lowered performance, despite using the same quantity of annotations and a much larger pool of unlabeled data. However, the effectiveness of semi-supervised methods have been well proven. For instance, SSDA3D [1] demonstrated that leveraging Waymo annotations in a semi-supervised manner significantly improves a 3D detector's performance on nuScenes when compared to using only the nuScenes training set.

**Contributions**
- Mismatch between claimed contribution and methodology: The proposed PPG appears to be a combination of pseudo-labeling and model ensembling to train the region proposal component. The model aggregates pseudo-labels from various detectors trained on different datasets. However, there seems to be a gap between this approach and the paper's main claimed contribution: transferring annotations across datasets with divergent protocols. As illustrated in Figure 1, these divergences include key challenges such as: (1) The same entity being assigned different category labels across datasets (e.g., "pedestrian" vs. "person"); (2) Different bounding box granularities for the same entity (e.g., annotating a "cyclist" and "bicycle" separately versus as a single "cyclist" entity). The paper lacks a clear discussion or evidence demonstrating how the proposed approach specifically resolves these two fundamental annotation conflicts. The authors need to explicitly present their method focusing on these challenges and show how it harmonizes or transfers such divergent annotations.
- Most importantly, **the differences between Domain Adaptation and the proposed label transferring task are not clear, as both tasks aim to use annotations from other datasets or domains. The paper offers no discussion or comparison.**

**References**

[1] Wang, Yan, et al. "Ssda3d: Semi-supervised domain adaptation for 3d object detection from point cloud." Proceedings of the AAAI Conference on Artificial Intelligence. Vol. 37. No. 3. 2023.

**Questions:**

- For the first row of Figure 1, I suggest the authors add category annotations near the bounding boxes to make the figure more self-contained.

---

### Official Review · Reviewer_tCdV · 2025-10-29

**Soundness:** 4
**Presentation:** 3
**Contribution:** 4
**Rating:** 8
**Confidence:** 5

**Summary:**

The paper addresses the issue of label space inconsistency across different object detection datasets. Unlike prior work, the proposed approach tackles both class-level and box-level inconsistencies. The method consists of two main components: a privileged proposal generator that leverages ground truth labels within each dataset, and a semantic feature fusion module that refines proposals in a class-aware manner.

**Strengths:**

- Addresses an important and underexplored problem in data-centric research.
- Effectively integrates class information into the transfer model, leading to large improvements on downstream tasks.
- Demonstrates strong generalization across different detection models (Faster R-CNN, RT-DETR, and YOLO, as shown in the appendix).
- Provides thorough ablations on downstream training strategies (Table 3) and class-aware attention (Table 4).
- Presents compelling qualitative results.

**Weaknesses:**

- No discussion or evaluation on synthetic object detection datasets—what is the reason for their exclusion?
- Missing comparison with Liao et al. (2024) or at least with the experimental setup/scenario presented in that work.

**Questions:**

- In Line 258 (“To maintain label … identical names”), does this mean that even if two datasets share the same class name, they are treated as distinct? If so, could this unintentionally encourage the model to differentiate datasets based on domain rather than class semantics (e.g., when the class “bicycle” is consistent across datasets A and B)?

---

### Official Review · Reviewer_GUUF · 2025-10-30

**Soundness:** 2
**Presentation:** 2
**Contribution:** 2
**Rating:** 2
**Confidence:** 4

**Summary:**

This paper aims to tackle multi-dataset object detection without requiring shared label taxonomies and consistent spatial annotations. To this end, the authors introduce Label-Aligned Transfer (LAT), a data-centric framework comprising a privileged proposal generator and a semantic feature fusion module. The key idea is to project the label space of source datasets to target dataset, thereby generating target-aligned pseudo annotations that facilitate downstream detector training. Experiments on multiple object detection benchmarks show that LAT achieves state-of-the-art results.

**Strengths:**

* The paper tackles a practical and challenging problem in multi-dataset object detection, where source and target datasets do not share the same label space and annotation styles.
* Label-aligned transfer is proposed to tackle label and annotation inconsistency between source and target datasets, which enables joint training on heterogeneous datasets.
* Evaluations on two label transfer settings demonstrate that the proposed approach achieves promising results.

**Weaknesses:**

* Unjustified claim regarding label taxonomy. The introduction mentions that the proposed method does not require shared taxonomies between source and target datasets. However, the benchmarks used in the paper still exhibit class overlap (e.g., common categories such as pedestrian and vehicle). Intuitively, this claim should be justified by using source and target datasets with non-overlapped object classes.
* The reported results in Table 1 are not entirely convincing. First, the authors mention that only 3,000 images are sampled for each dataset. Given that the nuImages dataset contains over 60,000 training images, the use of 3,000 samples may lead to biased results. Second, the comparison with Plain-DET is unfair. This paper adopts DINOv2 as backbone, whereas Plain-DET uses ResNet-50.
* Missing comparisons with a pseudo label baseline under RT-DETR setting. It is important to validate that the proposed method outperforms naïve pseudo label baseline.
* High computational cost. The proposed method adopts a multi-stage training pipeline that scales quadratically with the number of datasets. Specifically, for N datasets, the first stage trains N individual detectors, and the second stage requires each model to generate pseudo-labels on the remaining N−1 datasets, resulting in $O(N^2)$ inference passes. The approach may become prohibitively expensive as N grows, limiting its scalability in large-scale multi-dataset scenarios.
* It would be better to evaluate the method under a strict multi-dataset setting, where  the total training iteration remains the same when the number of datasets increases. (similar to Table 1 in Plain-DET).

**Questions:**

* Does the pseudo label baseline in Tables 2 and 3 incorporate filtering strategy? Table 8 shows that pseudo label with filtering significantly outperforms naïve pseudo label approach.

---

### Official Review · Reviewer_TBvP · 2025-10-31

**Soundness:** 3
**Presentation:** 3
**Contribution:** 3
**Rating:** 6
**Confidence:** 4

**Summary:**

Proposes Label Aligned Transfer (LAT), a method to translate object detection annotations with varying protocols to the label space of a target dataset. The method first fuses pseudo-labels with ground truth annotations via a "privileged proposal generator”, and then refines region features with a class-aware semantic feature fusion module. The proposed approach is shown to significantly improve performance on several benchmarks.

**Strengths:**

– Studies a challenging problem with real-world utility

– The paper is clear and well-organized, and the novelty over prior work is outlined clearly

– The experimental results validate the effectiveness of the proposed method

– The method ablations clearly justify the role of each component in improving performance

– The qualitative results included in the appendix are compelling

**Weaknesses:**

– LAT seems to match but not exceed the performance of LGPL – since code is unavailable, it would still be good to describe in more detail the distinctions between the two methods and relative pros/cons of each

– It would be good to also benchmark a frontier model (eg. SAM-2), to confirm that the label transfer task still has practical utility

– The paper would be strengthened by a more fine-grained performance analysis – which classes benefit/degrade the most from this type of transfer, and why (size, frequency, location, something else?)?

**Questions:**

Please address the weaknesses listed above, especially around comparison to LGPL and missing comparisons and analysis.

---

### Note · Authors · 2025-11-14

I have read and agree with the venue's withdrawal policy on behalf of myself and my co-authors.